# The mediating effects of public genomic knowledge in precision medicine implementation: A structural equation model approach

John Jules O. Mogaka [ID]*, Moses J. Chimbari

Department of Public Health Medicine, University of KwaZulu-Natal, Durban, South Africa

* johnmogaka2@gmail.com

**Data Availability Statement:** The datasets were deposited in Zenodo (Public data repository): "Structural equation model mediation analysis dataset" http://doi.org/10.5281/zenodo.3902899

## Abstract

Precision medicine emphasizes predictive, preventive and personalized treatment on the basis of information gleaned from personal genetic and environmental data. Its implementation at health systems level is regarded as multifactorial, involving variables associated with omics technologies, public genomic awareness and adoption tendencies for new medical technologies. However, interrelationships of the various factors and their synergy has not been sufficiently quantified. Based on a survey of 270 participants involved in the use of molecular tests (omics-based biomarkers, OBMs), this study examined how characteristics of omics biomarkers influence precision medicine implementation outcomes (ImO) through an intermediary factor, public genomic awareness (represented by User Response, UsR). A structural equation modelling (SEM) approach was applied to develop and test a 3 latent variable mediation model; each latent variable being measured by a set of indicators ranging between three and six. Mediation analysis results confirmed a partial mediation effect (an indirect effect represented as the product of paths 'a' and 'b' (a*b)) of 0.36 at 90% confidence level, CI = [0.03, 9.94]. Results from the individual mediation paths 'a' and 'b' however, showed that these effects were negative(a = -0.38, b = -0.94). Path 'a' represents the effect of characteristics of OBMs on the mediator, UsR; 'b' represents the effect of the mediator, UsR on implementation outcomes, ImO, holding OBMs constant. The results have both theoretical and practice implications for biomedical genomics research and clinical genomics, respectively. For instance, the results imply better ways have to be devised to more effectively engage the public in addressing extended family support for extended family cascade screening, especially for monogenic hereditary conditions like *BRCA*-related breast cancer and colorectal cancer in Lynch syndrome families. At basic biomedical research level, results suggest an integrated biomarker development pipeline, with early consideration of factors that may influence biomarker uptake. The results are also relevant at health systems level in indicating which factors should be addressed for successful.

**Funding:** The author(s) received no specific funding for this work.

**Competing interests:** The authors have declared that no competing interests exist.

## Introduction

Advances in genomic technologies have deepened insights into the complex structure and function of the human genome, with far-reaching implications in medicine and health care. The landmark Human Genome Project(HGP) [1, 2] took 13 years at a cost of over U\$2 billion to sequence the first human genome. The project's ripple effects prompted multidirectional breakthroughs in biotechnology, including the use of next generation sequencing (NGS) that has dramatically reduced the cost and turn-around of genome sequencing to just a day at cost ofU\$1000. Precision medicine (PM), viewed as a future paradigm for medicine [3, 4], stems from such bio-technological breakthroughs. It has potentials to usher in a more precise and targeted approach to disease screening, diagnosis and treatment by accentuating predictive, preventive and personalized medicine. With advances in omics technologies (tools used in the study of the genome), PM's achievements are no longer confined to biomedical research but are steadily moving to the clinical practice settings [5]. Early PM applications have resulted in improved genetic screening for newborns (for fatal yet preventable conditions) [6], oncology (genetic screening to avoid metastatic and aggressive cancers) [7, 8] and population pharmaco-genomics (in avoiding unnecessary pharmacological adverse reactions) [9, 10]. Biomarkers discovered through omics technologies are driver-factors in realizing the promise of precision medicine, at both individual and population health spaces. Yet, because of the complexities involved in integrating new biomarkers into clinical care, many institutions and health systems may face challenges, including resource inequities, differences in regulatory frameworks, differing social contexts, economic statuses and national health priorities. A PM implementation model may be crucial in capturing these complexities and point to better implementation pathways, especially at the health systems level.

Existing evidence in the field of implementation research offer a range of theoretical frameworks that explain multi-level factors which can influence PM implementation at systems level [11–15]. This consists of innovation-, individual- and institutional-level factors as illustrated in Fig 1. An early consideration of characteristics of the innovation that is to be adopted (i.e. omics biomarkers) in the discovery pipeline may help to emphasize tools, products, and

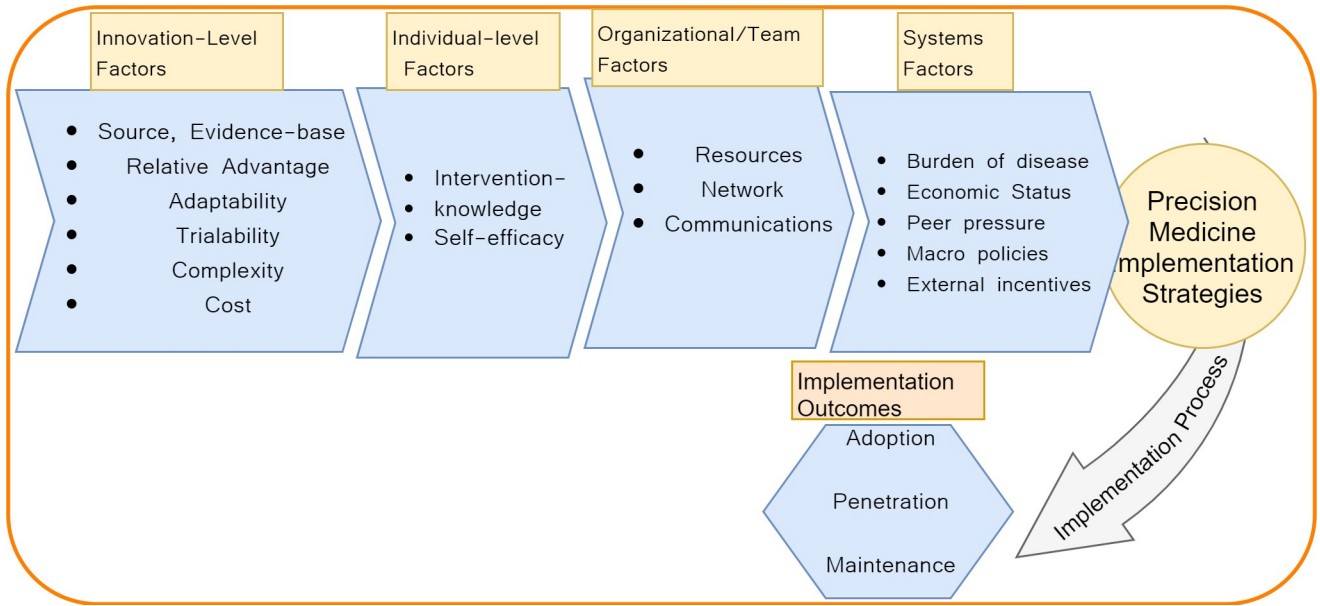

**Fig 1. A precision medicine implementation meta-theoretical framework.**

strategies that may mitigate variations in uptake not only across patient, provider, and/or organizational contexts, but also across time spaces. Furthermore, consideration of such individual and institutional factors may help identify gaps in uptake facilitation and make for early mitigation of the gaps across systems (e.g., in terms of resource allocation). On the other hand, early consideration of systems-level factors may help in addressing misaligned and non-friendly genomic policies.

Even though there exists a number of theoretical frameworks that may inform optimal implementation of PM at health systems level, little research specific to PM implementation has been done, particularly in quantifying these determinants. Moreover, little is understood about the mechanisms through which one set of factors transmit effects onto other related factors to achieve desired implementation outcomes, especially in resource constrained settings.

In this paper, we hypothesize that the relationship between characteristics attributable to omics biomarkers (OBMs) and their utility in clinical settings may not be simple. For instance, omics biomarkers, as the foundation of PM, have been shown to present clear advantages over traditional biomarkers, especially for treatable-if-diagnosed-early diseases but for which patients cannot benefit from existing treatment approaches if discovered at later stages of development., e.g., systemic amyloidosis [16]. This fact however, if considered in isolation with other factors, does not necessarily accord omics biomarkers ready integration into clinical application. Unique contextual factors likely influence the extent to which OBMs are integrated in routine clinical application. This study therefore sought to test the extent to which the relationship between omics biomarkers and their clinical uptake is intermediated by user response related to public genomic awareness or engagement (patients and providers). Fig 2 outlines the basic mediation model that is hypothesized and referred to in this study. In the simple mediation model suggested in Fig 2, it is hypothesized that the observed relation between characteristics of omics biomarkers (OBM), referred to as the exogenous (independent) variable, and PM implementation outcomes, referred to as the endogenous (dependent) variable, can be explained by the effect of user response from the public (patients and practitioners), a third factor referred to as the mediator. This third factor is triggered by public genomic awareness. It is further suggested that the observed indicators are caused by the three latent variables, hence the direction of the variable-indicator arrows; i.e., in this reflective model, the causal action flows from the latent variable to the indicators. The small residual error circles indicate error in measurement attributable to the indicators they point to.

Studying contextual effects by investigating mediating variables has the potential to extend the generalizability of PM implementation efforts to different settings. By analyzing mediation effects in this way, this study hopes to contribute to the refinement of existing implementation knowledge on PM.

In the following sections, the utility of examining mediation effects in PM implementation is presented. The results and discussion sections present effect size and other intermediation statistics. The "likert" ver.1.3.5 [17] and "lavaan" version 0.6–3 [18, 19] in R version 3.6.0. [20] were used for descriptive and inferential statistics, respectively. This was done using structural equation modelling (SEM), a method that estimates all parameters simultaneously and generally results in unbiased estimates [21].

## Theoretical background, latent variable measurement and hypothesis statements

Biomarker discovery might be the primary focus of most biomedical research, but the long-term goal for PM is to fully integrate them into the healthcare delivery system to enhance quality of medical care through improved disease screening, diagnostics and therapeutics. Apart

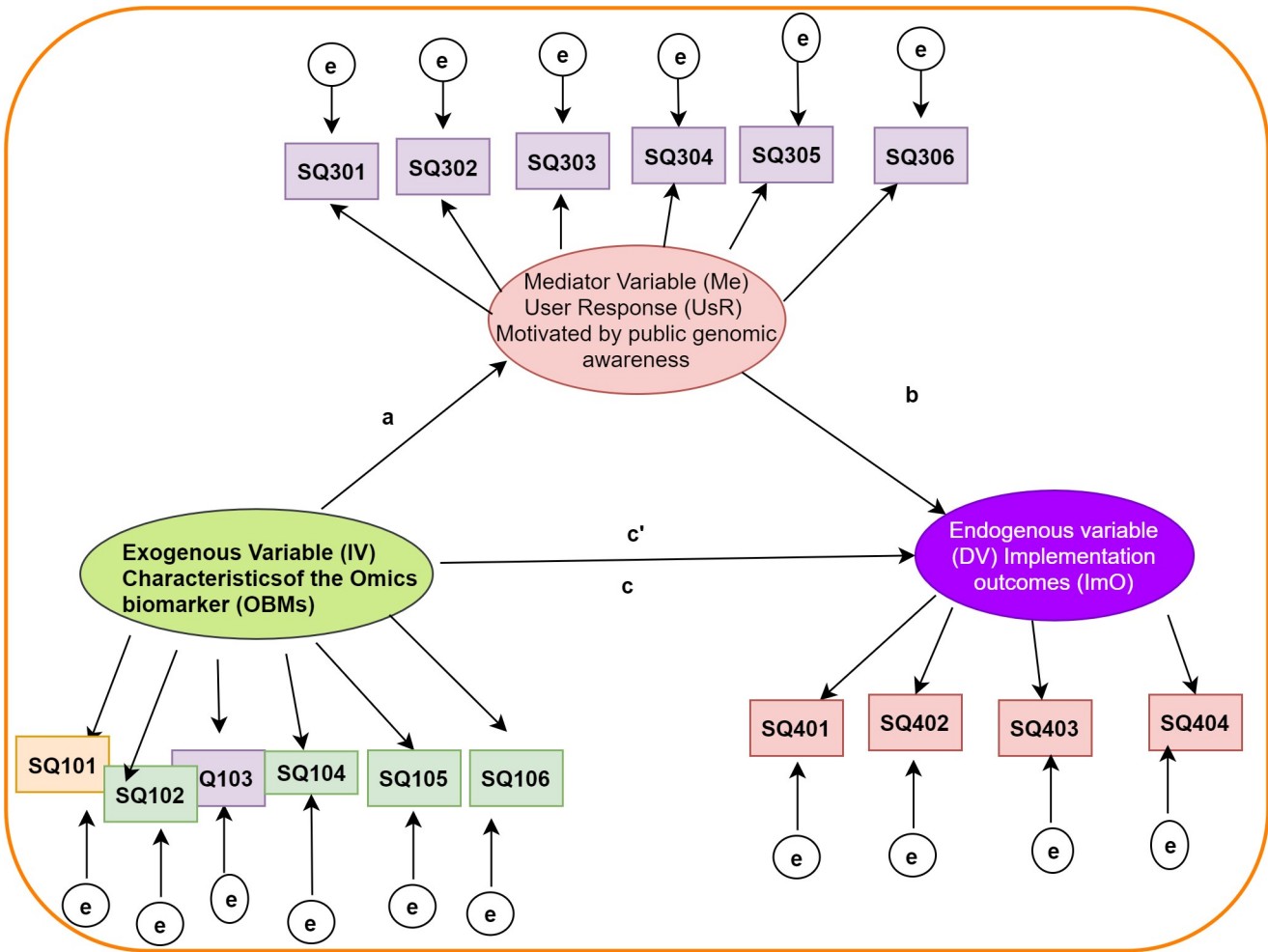

**Fig 2. Hypothesized systems level precision medicine implementation mediation model.** Key: Large oval shapes = latent (unobservable) factors; Rectangles = indicator (observable) variables; Arrows = hypothesised correlation direction; Small circles = residual errors explaining measurements errors; Me = the mediator variable, a = the effect size of the independent variable on the mediator, b = the effect of the mediator on the dependent variable controlling for X, and c' = the direct effect of X on Y controlling for Me.

from innovation-level factors, regulatory, social, technical and other contextual factors need to be attended to in order to realize the broader implementation and full potential of precision medicine. Three factors, hypothesized to influence PM implementation in this study, are briefly discussed below. They are innovation-level factors associated with omics biomarkers (e.g. sufficient evidence generated to validate biomarkers), individual-level factors (i.e. public genomic awareness through patient and provider engagement) and system level factors that indicate institutionalized uptake of omics-based biomarkers.

## Characteristics of omics biomarkers (OBM)

Years of sustained developments in omics technologies and expanded knowledge of disease pathogenesis at the molecular level have resulted in novel biomarkers useful for disease characterization, early diagnosis, and drug discovery and development [22–24]. The biomarkers help to identify causative gene mutations or polymorphisms of susceptibility and can also reveal DNA and RNA characteristics related to drug responses. Even though single gene biomarkers have existed for a long time (e.g. in linking genetic effects for both patient and family in cystic

fibrosis testing and to monitor particular effects on large populations e.g. HIV mRNA, HCV mRNA), recent developments have led to an expanded array of omics biomarkers with improved diagnosis, characterization, and therapy selection [25]. Omics biomarkers are genomic characteristics that indicate normal biologic processes, pathogenic processes, and/or response to therapeutic or other pharmacological interventions. They include single nucleotide polymorphisms (SNPs), DNA modification, e.g. methylation, insertions and deletions (Indels), RNA sequences, RNA expression levels and microRNA levels. Successful biomarkers offer a span of benefits including patient stratification for preventive interventions [26], screening populations for early disease detection [27], subtyping disease to facilitate chemotherapy tailored at the molecular level [28], and monitoring response to treatment [29]. Despite these apparent advantages, there are varying perceptions about the effectiveness of omics biomarkers.

Single gene mutation testing to diagnose or determine predispositions to certain disorders is common in clinical settings. Although polygenic diseases are more common than single-gene disorders, clinical tests using molecular biomarkers for polygenic (multifactorial) disorders are not routinely done. The multifactorial nature of polygenic disorders presents challenges in the discovery of appropriate omics biomarkers. Due in part to high population prevalence of most polygenic disorders, lack of clear Mendelian transmission patterns and phenotypic heterogeneity associated with these diseases, the validity and clinical utility of some biomarkers may vary based on specific population characteristics. Such heterogeneity has implications on case and control populations for such biomarkers. Moreover, quality biospecimen and right bio-sample quantities maybe prerequisites for biomarker discovery research for well-defined clinical applications to be ascertained. Therefore, characteristics associated with biomarkers influence perceptions about their advantages, clinical validity and public health applicability.

## Involved individuals' response to omics biomarkers (UsR)

The use of biomarkers for diagnostics, prognostic or predictive purposes is beneficial in circumstances such as identifying inherited susceptibility for future disease, thereby informing tailored and timely preventive strategies, besides providing means for optimizing drug therapies based on individual pharmacological responses.

However, use of genetic profiling is often met with anticipation, skepticism and concern at personal level [30, 31]. Genetic testing necessarily demands much effort in anticipating, understanding and addressing associated ethical, legal and social implications. Use of omics biomarkers, therefore, presents particular challenges with respect to providers' ethical and professional responsibilities, including the appropriate use of genomic information in health care settings. For example, women found to have an inherited susceptibility to cancer after a genetic test on them or their relatives might face social discrimination or stigmatization. In addition, family members may be disenfranchised by the very process of genetic testing, particularly if some members wished to pursue testing and others did not, or if some individuals found out information about their own risk through genetic test results of other family members, or if those with normal test results experienced survivor guilt. Genetic test results could also be misinterpreted [32] or create an impetus toward the use of unproven medical therapies due to despair [33]. Yet some genetic tests do not generate these concerns. An excellent example for this is the case of neonatal and fetal omics/genetic screening exercises to determine conditions amenable to early interventions. While the differences between desirability and skepticism for testing in the above scenarios may seem intuitively clear, many aspects of testing; nature of test, mode of inheritance, person tested, social or medical context–might

contribute to their acceptability or rejection among involved individuals, both as providers and as patients.

## Precision medicine implementation outcomes (ImO)

The hypothesized implementation model explains what influences PM implementation outcomes. Such evidence can inform the design and execution of implementation strategies that aim to change relevant determinants. The model explicitly depicts determinants that influence implementation outcomes as nonlinear, considering individual barriers and enablers that may interact in various ways within and across levels.

Indicators for the implantation outcome construct were drawn from the RE-AIM Framework [34]. Implementation outcomes are concerned with the evaluative dispositions of implementation efforts. Using elements of RE-AIM, those in research or practice can make use of necessary information to justify adoption of the biomarker, and how to maintain it if adopted or widen its reach (penetration) into a given service setting.

**Hypotheses.**   In formulating the following hypotheses we followed the segmentation and transmittal approaches as expounded in Rungtusanatham et al. [35].

H1. Characteristics of an omics-based biomarker (OBM) has a positive effect on user response (public genomic acceptance).

H2. User response due to public genomic awareness mediates the relationship between characteristic of an omics-based biomarker and precision medicine implementation outcomes.

H3. The effect of user response on implementation outcomes is statistically significant.

## Methods

### Study participants and procedure

This study was approved, and institutional review board permit obtained from the University of KwaZulu-Natal (BREC Permit Ref No BE513/18).

Snowball sampling method was applied in identifying potential participants from population of interest for this study. The seed (initial) sample population composed of individuals affiliated to various academic institutions and organizations known to be involved in molecular/genetic testing and omics-based biomarkers across Africa. Other participants were identified in precision medicine-related academic conferences and invited to participate. Guided by general principles of the Nuremberg Code, the Declaration of Helsinki and institutional review board as already noted, the study package was distributed via email to potential participants between June and July 2019. Since this study contained negligible risk of potential embarrassment or other ethical dilemmas that are usually associated with snowball sampling in many other studies, initial participants were encouraged to forward the email containing the study package to their colleagues. The study package included an invitation letter with study description, consent form and a link to an online questionnaire. We hosted the questionnaire online on a platform hosted and supported by Optimal Workshop [36]. Besides offering convenience to study participants, the online platform ensured data security and confidentiality. Participants were encouraged to complete the study but informed that the platform was open only for a period of 60 days (1$^{st}$ June to 31$^{st}$ July). Snowball sampling was deemed the ideal sampling method for the study as it was expected that initial study participants would likely know others in the same industry or academic circles as themselves and hence, could collegially inform others about the study and its potential benefits. However, despite this advantage, use of this

method meant that it was not possible to determine the sampling error based on the obtained sample.

Online means of data collection are known to be convenient and cost-effective in reaching large numbers of participants over a relatively short periods of time as compared to conventional paper-based surveys [37]. However, this approach is prone to increased risk of survey attrition—participants dropping out. To address this risk and possibly curb the potential havoc such participant attrition may have on our study findings, we set crucial question settings on the online survey tool to 'compulsory'. This implied that participants had to answer the first survey question to proceed to the next one and all questions had to be completed before successfully submitting the survey form at the finish line. However, the survey platform still recorded any attempt to participate, completed or not. This measure meant that attempt by quitters to participate was registered. Unfortunately, this measure did not discriminate between genuine quitters and subsequent attempters. To eliminate possibility of a bias due to survey attrition or attrition affecting study findings and thus having a negative statistical implication on our model, we only used data from participants who actually completed the survey for all statistical analyses.

We operationalized "members of academia" as consisting of those involved in biomedical research related to translating newly discovered molecular biomarkers (OBMs) for purposes of clinical or population health use; members of "industry" as those involved in the clinical use of the biomarkers (e.g., clinical pharmacogenomic testing); or those involved in commercial entities related to OBMs (e.g., Direct-To-Consumers Genetic Testing, DTC-GT); "precision medicine implementation" was defined as constituting the process of translating newly discovered omics-based biomarkers (OBMs) for purposes of clinical or population health use; OBMs were taken to be either candidate genetic biomarkers that are in the process of being clinically validated, or those that are already validated and in clinical use. Additional participation eligibility criteria included working on Africa-based precision or genomic medicine projects i.e., study participants who are primarily in continental Africa.

Participants were asked to answer four short sections in the questionnaire. The first three sections related to the three factors (constructs) thought to influence PM implementation. The fourth section was designed to elicit demographic information about participants, including their age, gender and organizational affiliations. Regarding factors influencing PM implementation, participants were asked to rate their considered opinion on a five point "strongly agree" to "strongly disagree" Likert-type scale. The items that participants were responding to are listed in Table 1. Their responses formed the dataset underlying the analysis in this paper. As described in the Results section, differences in participant responses in relation to the organization type they belonged to was extensively explored and reported upon.

**Measures.** The measurement tool was earlier developed in a pretest study; its validation was presented in a separate paper that is related to the present study. A pretest study was carried out to test and validate the data collection tool and assess whether the proposed methods of data collection and analysis would meet study objectives. This was done to ensure appropriate domain sampling, good factor structure and high internal consistency. The pretest study was carried out with 31 subject matter experts (SMEs) that were not selected for participation in the mainstream data collection phase. From the pretest study, good scale score reliability (internal measurement consistency) for the entire tool was confirmed for the study population (omega ($\omega_t$) = 0.95). An omega measurement consistency coefficient of 0.95 indicated a high and accurate approximation of the tool's reliability [38]. In confirmatory factor analysis (CFA), satisfactory values of the fit indices were obtained (comparative fit index, CFI = 0.98; Tucker-Lewis Index, TLI = 0.97; root mean square error of approximation, RMSEA = 0.06; standardized root mean square residual, SRMR = 0.2). The tool was found to have an

**Table 1. Latent variable indicators.**

| | |
|---|---|
| **SQ101** | It is easy to obtain the right quantity of bio-samples to assure accuracy in biomarker test results |
| **SQ102** | It is easy to obtain specified quality of bio-samples to ensure accuracy in the biomarker |
| **SQ103** | The bio-marker has previously been tested among people with similar characteristics as the present target population |
| **SQ104** | Genetic counselling is part of the procedures when undertaking testing using this bio-marker |
| **SQ105** | The turn-around time for obtaining results after the genetic/omics biomarker test is reasonable for the intended use. |
| **SQ106** | There are step-by-step instructions on how to obtain samples from individuals for biomarker tests. |
| **SQ301** | Participants easily give consent to obtain samples from them for the purpose of biomarker testing |
| **SQ302** | Getting buy-in from the public (patients, and/or providers) in carrying out the biomarker testing is easy |
| **SQ303** | Publicity and free information publicly available about the genetic biomarker make potential users to willingly ask for the biomarker test |
| **SQ304** | Using this genetic/omics test has been regarded by most practitioners as an appropriate mechanism for patient management (e.g. aid in drug dosage decisions, in carrying fetuses to term or carry out prophylactic surgery). |
| **SQ305** | There is a considerable 'pushback' from practitioners as they feel the genetic/omics test is not consistent with their skills, role, or job expectations. |
| **SQ306** | Targeted individuals feel that the genetic test is in line with their family members' wishes, desires and expectations |
| **SQ401** | The genetic test is yet to be used as a routine practice within its intended service settings |
| **SQ402** | Practitioners are more willing to order the genetic/omics test more often whenever they deem it necessary to do so |
| **SQ403** | The number of eligible persons able to access the genetic/omics test is far less than the total number potentially in need of the service |
| **SQ404** | So far, the authorities that are supposed to acquire the biomarker testing service have communicated a decision to fully fund its roll out |

appropriate level of validity and reliability. Measurement indicators for the three factors are listed in Table 1. Indicators SQ101 to SQ106 for the exogenous variable "OBM" and SQ301-SQ306 and SQ401-SQ404 for the endogenous variables "UsR" and "ImO" respectively.

## Data analysis

The latent variable mediation analysis model under consideration in this paper has each of its 3 latent variables measured by a set of indicators: SQ101 to SQ106 for the exogenous variable "OBM", SQ301-SQ306 and SQ401-SQ404 for the endogenous variables "UsR" and "ImO" respectively. The indirect effect, the product of coefficients a*b as presented in Fig 2 and Table 2 corresponds to the effect of the independent latent variable "OBM" on the outcome variable, "ImO", through the mediator "UsR". The mediator is the process that explains why changes in the independent variable might result in changes in the outcome.

The "likert" package ver.1.3.5 [17], an R package designed to help in analyzing and visualizing Likert-type items, was used to provide descriptive statistics and summarize the Likert type responses. Data was then subjected to statistical analysis using "lavaan" (acronym for **la**tent **va**riable **an**alysis) version 0.6–3 [18, 19] in R version 3.6.0 [20]. The package was chosen for its collection of tools that can be used to explore, estimate, and understand a wide family of latent variable models, including factor analysis, structural equation, longitudinal, multilevel, latent class, item response, and missing data models. We applied structural equation modeling (SEM) principles, using a conceptual model, path analysis and a system of linked regression-style equations to capture relationships within a web of observed and unobserved variables. We estimated the relationships among the three latent variables, as well as tested the overall

**Table 2. Parameter estimates for fitted mediation model.**

| lhs | op | rhs | label | est | se | ci.lower | ci.upper | std.lv | std.all | std.nox |
|-----|-----|-----|-------|-----|-----|----------|----------|--------|---------|---------|
| UsR | ~ | OBM | a | -0.382 | 0.492 | -2.722 | -0.084 | -0.105 | -0.105 | -0.105 |
| ImO | ~ | OBM | c | -1.885 | 1.668 | -56.230 | -1.004 | -0.380 | -0.380 | -0.380 |
| ImO | ~ | UsR | b | -0.945 | 1.430 | -4.682 | -0.283 | -0.694 | -0.694 | -0.694 |
| direct | := | c | direct | -1.885 | 1.668 | -56.230 | -1.004 | -0.380 | -0.380 | -0.380 |
| indirect | := | a*b | indirect | 0.361 | 1.300 | 0.030 | 9.944 | 0.073 | 0.073 | 0.073 |
| total | := | c+(a*b) | total | -1.523 | 0.744 | -6.266 | -0.924 | -0.307 | -0.307 | -0.307 |
| UsR | $r^2$ | UsR | | 0.011 | | | | | | |
| ImO | $r^2$ | ImO | | 0.571 | | | | | | |

Key: lhs and rhs = Left and right hand side(of the model equation); op = operator (e.g., ~ = 'regression operator); est = unstandardized estimates; $r^2$ = r squared; ci.lower and ci.upper = lower and upper confidence intervals at 90% confidence level; Std.lv = only the latent and not the observed variables are standardized; Std.all = fully standardized solution (both latent and observed) variables are standardized to have a variance of one. std.nox = estimates in which the latent variables and endogenous observed variables are standardized but the exogenous observed variables are left in their raw scale, i.e. partially standardized estimates.

structural model in addition to individual paths. SEM was used to obtain effect sizes simultaneously from the exogenous variable to the mediator and mediator to the outcome variable, as well as the combined mediation effect, corrected for any attenuating effects of the measurement error (residual errors).

Although the Baron & Kenny method for testing mediation variables [39] is popular as a normal theory (NT) approach, we applied the alternative approach described by Shrout and Bolger [40] based on bootstrap data resampling procedures to establish confidence intervals for testing the statistical significance of our indirect effect. Standard errors (SE) were also bootstrapped. Bootstrap methods treat the collected research sample as a "population reservoir" from which a large number of random samples are drawn with continuous replacement such that the probability of selection for any given case remains equal over every random draw [40]. We requested 5,000 bootstrap samples, drawn by default with replacement from the full data set of 270 cases (our empirical sample) at 90% confidence intervals. We used the maximum likelihood (ML) estimation method. Although ML estimation method is usually good for continuous variables, it has been observed that ordinal variables with many categories, such as 5-point Likert-type scales of agreement, are usually safely treated as "continuous" [41].

The bias-corrected (but not accelerated) confidence interval method was selected. Estimates of indirect, direct, and total effects, 'a', 'b', and 'c' path coefficients and other parameters were requested for through the "lavaan" "parameterEstimates" function. The "parameterEstimates" function estimates the bootstrap parameters and extracts not only the values of the estimated parameters, but also the standard errors, the z-values, the standardized parameter values.

**Multigroup analysis.** Even though factors hypothesized to affect PM implementation in this study may seem to be premised on a multilevel framework (e.g. innovation-level factors, individual-level factors, and organizational factors, etc.), based on the available data, a multilevel structural equation modeling was not applicable. Previous research has shown that multilevel structural equation modeling is appropriate in handling clustered or grouped multivariate data; it demonstrates how levels of the within-group endogenous and exogenous variables vary over between-group units, hence explaining between-group variation of within-group variables [42]. Our data however, contained latent variables and indicators that only varied between units (study participants) and therefore lacked nested clusters, i.e., it lacked variables measured at different levels of sampling hierarchy.

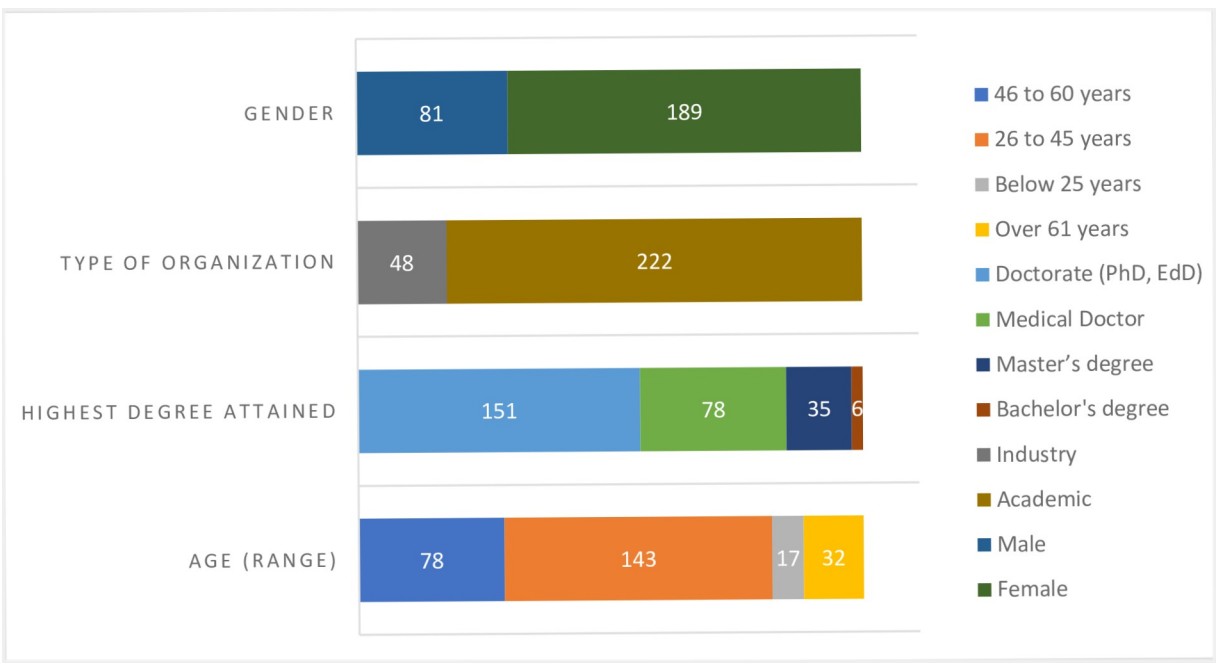

**Fig 3. Demographic summary of study participants.**

## Results and discussion

### Demographic characteristics of participants

Of a total of 442 who showed an initial attempt to complete the survey as registered on the study's online platform, 270 (61%) participants went ahead to successfully complete the survey. This implies that 172 attempters probably "abandoned" the survey.

Majority of participants belonged to the 26-45-year age bracket (>52%), whereas there were more males (70%) than females (30%). There were more participants affiliated to academic institutions (>82%) than industry (<18%) in the survey. A summary of basic demographic data of study participants is presented in Fig 3.

Descriptive statistics that were obtained for the data included linearity and multivariate normality evaluated for the 270 participants with complete data using R package 'psych' version 1.8.12 [43]. The squared Mahalanobis distance was plotted against quantiles of the chi-square distribution to detect detect outliers in multivariate data, as shown in Fig 4. Because

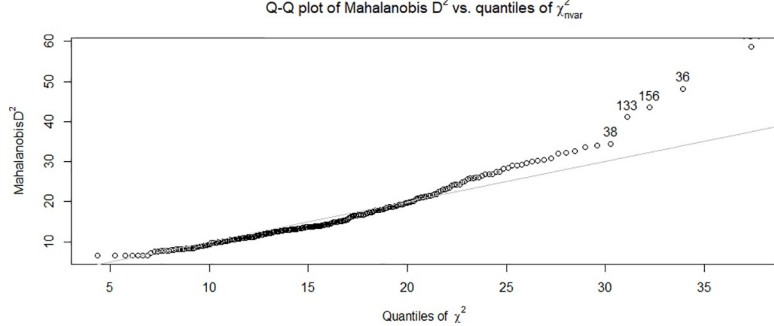

**Fig 4.** Quantile-Quantile (Q-Q) plot describing squared Mahalanobis distance (y-axis) against the quantiles of the chi-square distribution (x-axis).

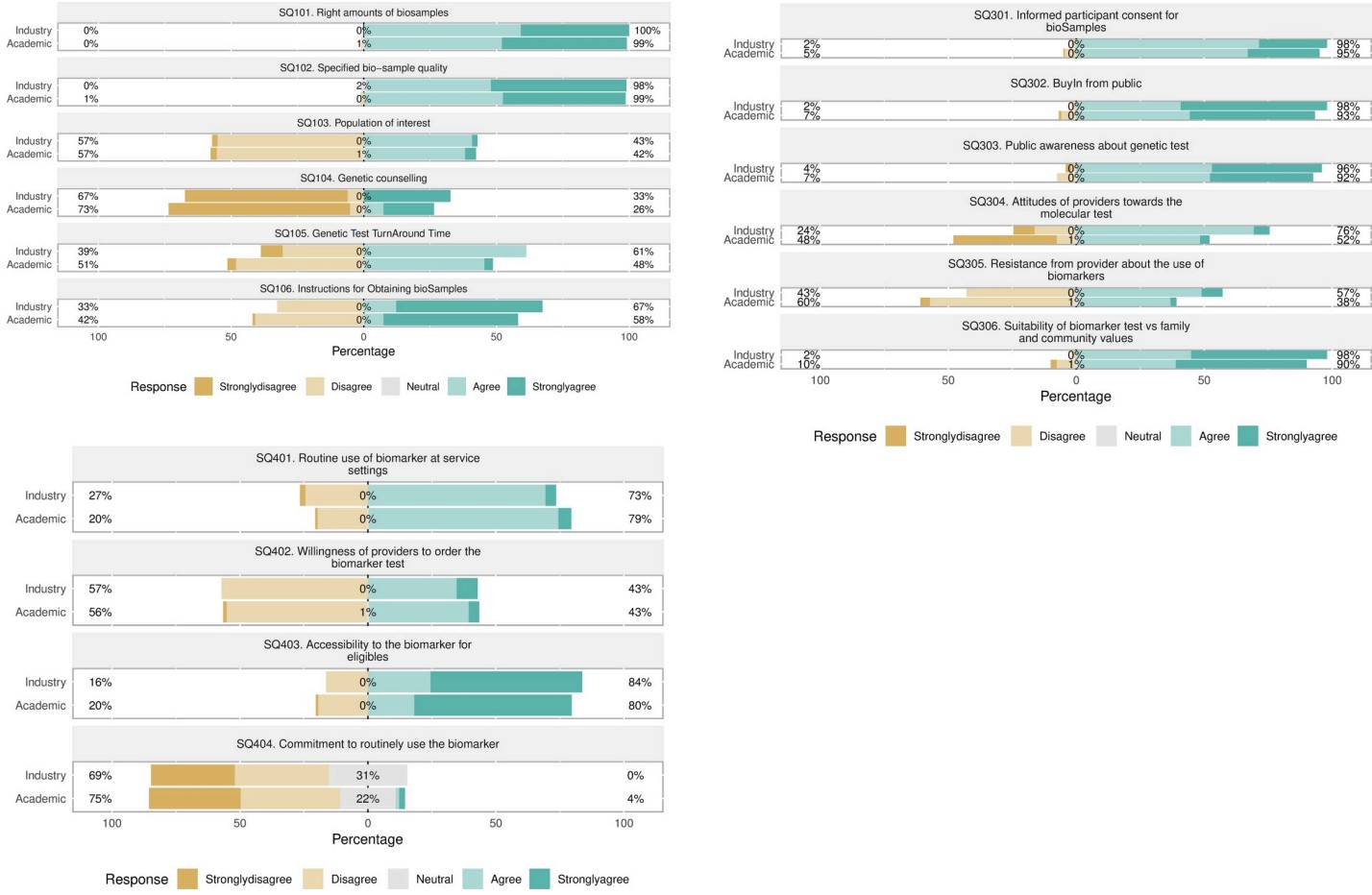

**Fig 5.** A. Survey responses to questionnaire section on characteristics of omics biomarkers. B. Survey responses on public genomic awareness presented as "user response"(UsR). C. Diverging stacked bar-charts for survey responses on implementation outcomes (ImO) construct.

points in the plot tended to fall along a straight line, suggesting that the squared Mahalanobis distance has an approximate chi-squared distribution. We therefore concluded that the data were distributed as multivariate normal (MVN). Despite one item being an outlier, its inclusion in the data set did not alter results, and therefore all items were included for further analysis.

**Descriptive statistics.** Fig 5A through 5c are diverging two-way bar-charts that graphically present the responses obtained from study participants. The charts show comparisons in percentages within subgroups of the survey population. All bars have equal vertical thickness, although panel heights are proportional to the number of bars in each graph. The x-axis labels are displayed with positive numbers on either side of the center reference point (0). The bars are horizontal to conveniently accommodate group and indicator labels as displayed horizontally on the y-axis. Each indicator is mapped onto a pair of stacked bar-chart. Responses correspond to each indicator of the three constructs as presented in the measures section of this paper. "OBM", "UsR" each has 6 by 2 panels while "ImO" has 4 by 2. Each indicator is responded to by two groups as indicated by group affiliation (academic or industry). Responses in percentages of each respondent subgroup who agreed with the indicator statement are shown to the right; the percentages who disagree are shown to the left; the center indicates those in neutral (neither agreeing nor disagreeing), all adding up to 100%. For instance, in

Fig 5A, there were no "strongly disagree" and "disagree" responses for the item "SQ101: It is easy to obtain the right quantity of bio-samples to assure accuracy in biomarker test results" among the "academic" subgroup (hence 0% at the far right of the stacked bar-chart corresponding to the item. Neither was there any "Neutral" response for this item. All participants (100%) in the academic subgroup agreed to the same statement, though approximately 60% of them "agreed" while 40% "strongly agreed". On the other hand, among the "Industry" subgroup, whereas there were none (0%) disagreements on item "SQ101", 1% of them were "Neutral". This means 99% of this subgroup agreed with the SQ101 statement. Each portion of the bar has a different color: left side is brown while right side is green, with a grey center. Darker colors indicate stronger agreement/stronger disagreement. The bar for the neutral position is split, half to the left side of the vertical zero reference line and half to the right side.

Fig 5A presents the survey responses on characteristics of omics biomarkers. As shown in the bar-chart, both participant subgroups gave similar agreeing responses to the first two items, while equally agreeing and disagreeing on the third item seeking their opinion on "Right number of bio-samples", "Specified bio-sample quality" and "Population of interest", respectively. Given the extreme care that is taken in ensuring accuracy of biomarker tests, seeking right amounts and qualities of bio-samples is necessary. Therefore, the response pattern to the first two items was expected. However, the responses given for the next four items ranging from SQ103 to SQ106 were surprising. The responses imply that on average, characteristics of population of interest and genetic counselling given to participants before obtaining bio-samples from them are generally not attended to with respect to biomarker testing among patients or study participants. On the other hand, genetic test results turnaround times seem not to be a serious concern among those in industry (e.g., practitioners) as compared to those in academia, as evidenced by the 12% disagreement margin between the two subgroups. 33% and 42% of those affiliated to industry and academia respectively disagreed with the statement "SQ106: There are step-by-step instructions on how to obtain samples from individuals for biomarker tests". This implies that there is more caution in handling bio-samples from patients/participants in "industry" settings (e.g., obtaining bio-samples from patients in clinical settings) than in academic biomedical research settings. Further analysis of these responses is beyond the scope of this paper but are presented in another paper related to this study.

Fig 5B presents survey responses on public genomic awareness as represented as "user response"(UsR). As shown in the bar-chart, as compared to industry-affiliated participants, slightly more academic-affiliated participants disagreed with the statement "SQ301: Participants easily give consent to obtain samples from them for the purpose of biomarker testing". This was expected because people are more likely to hesitate in giving consent for bio-samples in academic research settings than they would in clinical settings. Same expectation was expected in responses to "SQ306: Targeted individuals feel that the genetic test is in line with their family members' wishes, desires and expectations". In this case, more in academia than in industry (difference of 8%) disagreed with this item. The rest of the responses to items in this construct can be seen in Fig 5B. Each panel of the plot shows a breakdown of the responses into categories defined by the criterion listed in its left strip label and the legend at the bottom of the plot.

Fig 5C presents survey responses on the construct "Implementation outcomes (ImO)". As shown, out of the four items measuring this construct, the item "SQ404: So far, the authorities that are supposed to acquire the biomarker testing service have communicated a decision to fully fund its roll out" received the greatest "Neutral" responses (31% and 22% among industry and academia respectively). This may be reflective of the relatively long time it takes for those in health system authority to communicate decisions on adoption of new biomarkers for routine use. The rest of participant responses are presented on "ImO" are presented in Fig 5C.

**Inferential statistics.** The goal of biotechnological advances is to effect improved health outcomes. This study sought to test what effects perceptions about characteristics of omics-based biomarkers, also referred to as molecular tests and/or genetic biomarkers, may be having on precision medicine implementation outcomes. The study also sought to explain the process through which such effects occur, i.e. the mediation effect of public genomic awareness (PGA) on the relationship between "OBM" and "ImO". In this section, we test a set of three hypotheses whose outcome logically infers these correlational pathways. Statistical significance of effect sizes is used to infer theoretical and practical importance of the effects.

Fig 6 represent the latent variable mediation model with the three latent variables: "OBM", "UsR" and "ImO", each measured by a set of observable indicators as shown. The model shows the relationships of characteristics of omics-based biomarkers (OBM), public genomic awareness represented by user response (UsR) and precision medicine implementation outcomes (ImO). Path 'a' is the coefficient for the exogenous variable "OBM" as it effects mediator "UsR". Paths 'b' and 'c' are the coefficients in the model predicting the dependent variable "ImO" from both "UsR" and "OBM", respectively. Path 'c' quantifies the direct effect of "OBM", whereas the product of (a*b) quantifies the indirect effect of "OBM" on "ImO" through "UsR", while tracing the effect of "OBM" on the outcome variable, through the mediator. The effect sizes are based on standardized model parameters. For instance, the indirect effect is interpreted as the amount by which two cases that differ by one unit on "OBM" are expected to differ on "ImO" through the effect of "OBM" on "UsR", which in turn affects "ImO". In other words, the mediator explains why changes in the independent variable might result in changes in the outcome. The direct effect is interpreted as the part of the effect of 'OBM' on 'ImO' that is independent of the pathway through 'UsR'.

The mediation model was specified using appropriate syntax in lavaan version 0.6–3 [18, 19] (the r code is available). However, the model's modification indices (MI) suggested modification paths to improve the fit of the model. Almost all the modification indices were considered but others were not implemented owing to conflict with theory informing the model. The mediation model was then updated.

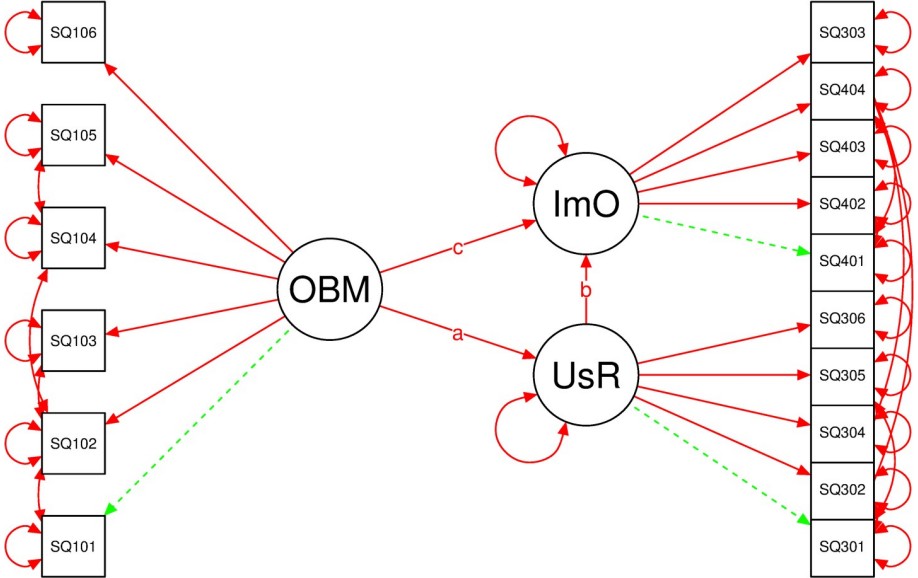

**Fig 6. Structural model showing the relationship between omics-based biomarkers (OBM), public genomic awareness represented by user response (UsR) and precision medicine implementation outcomes (ImO).** Red lines indicate estimated parameters while green lines indicate fixed parameters.

Examination of ANOVA results after testing the overall fit differences between the two models (the updated and initial model) indicated an improved updated model with a lower chi square value(11873.71 (improved) vs 11907.05 (initial model). The updated model was also better fitting than the initial model as indicated by a much lower Akaike Information Criterion (AIC = 91 vs 92) and Bayesian Information Criteria (BIC = 11711.79 vs 11748.72). Therefore, the updated model was used for mediation analysis and for hypothesis testing.

Since the product (a*b) in mediation analysis is often non-normally distributed as explained in [44], we used the bootstrap methodology [45] for more accurate standard errors and confidence intervals. Generally, bootstrapped confidence intervals and standard errors are more accurate for significance testing because they do not depend on an assumption of normality [45]. The summarized parameter outputs for the updated fitted mediation model are presented in Table 2. The measurement model showed an acceptable fit according to simple SEM fit statistics and indices: Root Mean Square Error of Approximation (RMSEA) = 0.056; Standardized root mean square residual (SRMR) = 0.070. Rule of thumb guidelines are that RMSEA ≤0.06 and SRMR ≤ 0.08 for acceptable and/or good fitting models [46].

Labels 'a', 'b', and 'c' are regression weights (as also illustrated in Fig 4). The 90% CI for (a*b) was obtained by a bias-corrected bootstrap, with 5,000 resamples.

**Hypotheses interpretation.** To appropriate sample effect sizes to the general population, bootstrap confidence intervals for the effect sizes were used. Confidence intervals (CIs) are often recommended for effect size interpretation [47] as they show the range within which the true population effect is likely to lie. Therefore, we used confidence intervals, rather than p values, in ascertaining statistical significance of effect sizes. We used bias-corrected bootstrap confidence intervals based on 5000 bootstrap samples at 90% confidence level. To find a meaningful and comparable scale, and because their coefficients are standardized, effect sizes are interpreted in standard deviation units.

*H1. Characteristics of an omics-based biomarker (OBM) affect user response (UsR)*

From Table 2, the effect size represented by path 'a' corresponds to the effect size of perceptions on characteristics of OBM on the mediating variable, user response (UsR). Given that the standardized regression coefficient can be used as an effect size measure for the 'a' coefficient [48], and that a correlation coefficient can be interpreted as the number of standard deviations that the dependent variable is expected to increase for a change of one standard deviation in the independent variable:

From Table 2, a = -0.105 (90% CI [-2.7217556, -0.0843599]).

Since the lower and upper confidence interval bounds do not contain zero, we can safely conclude that the influence (effect size) of characteristics of omics biomarkers (OBM) on user response (UsR) is non-zero in magnitude at 90% confidence level (0.1 SL). Also, since the confidence interval does not contain the null hypothesis value, we therefore reject the null hypothesis that the true influence of OBM on UsR (the 'a' effect size) is zero at 0.1 level of significance, in favor of the alternative hypothesis that characteristics of an omics-based biomarker (OBM) affect user response (UsR).

The point estimate of -0.105 corresponding to the value of standardized 'a' coefficient implies that for every (one) upward standard deviation change in characteristics of omics biomarker (OBM) in the population, there is a corresponding decrease of 0.105 standard deviations in user response (UsR) controlling for implementation outcomes (ImO) in the model. This is surprising given the expectation that as the public (patients and providers) become more aware of characteristics of omics biomarkers, their response should be positive or more favorable. But this expectation is not supported by the data from our sample.

The most probable explanation for this kind of negative observation could be related to the current dilemma in precision medicine of balancing the ever-advancing biotechnology with an appropriate evidence threshold for moving promising technology from research to practice. In part, initial omics technology discoveries have fueled increased expectations of major break-throughs in medicine. However, as deeper insights are uncovered about genetic variations, their interactions and products, a disconcerting mismatch between expectations and reality sets in. There are few diagnostic and screening tests based on individual genetic makeup, disease biomarkers and other genomics applications with proven clinical utility, e.g. HLA-B* 5701 (used in pharmacogenomics tests before starting HIV patients on abacavir to reduce the risk of hypersensitivity reaction) [49], and HFE testing (screening asymptomatic persons for HFE mutations) [50]. On the other hand, even though there are many such promising genomics applications (OBMs), most lack sufficient clinical utility evidence to support their routine use in clinical practice or population screening [51].

### H2. User response (UsR) through public genomic engagement affects precision medicine implementation outcomes (ImO)

From Table 2, the path representing 'b' codes the relation between the mediating variable (UsR) and the outcome variable (ImO) adjusted for the independent variable (OBM).

From Table 2: b = -0.694 (90% CI [-4.68, -0.283]).

Just as coefficient 'a' above, the lower and upper confidence interval bounds for coefficient 'b' do not contain zero, hence we can safely conclude that the effect size of user response (UsR) on implementation outcomes (ImO) is non-zero in magnitude (confidence level = 90%). Given that the confidence interval does not contain the null hypothesis value of 0, we reject the null hypothesis that the true population influence of UsR on ImO ('b') is zero (SL = 0.1). We therefore accept the alternative hypothesis that user response (UsR) through public genomic awareness significantly affects precision medicine implementation outcomes (ImO).

The point estimate of 'b' is negative at -0.694. This implies that for every standard deviation change in 'user response' in the population, there is a corresponding decrease of 0.694 standard deviations in implementation outcomes (ImO), controlling for characteristics of omics biomarkers in the model. Once again, the expectation is that more public genomic knowledge should result in an upsurge of omics biomarker uptake (more implementation outcomes). A possible reason for this surprise finding could still be linked to perceived quality of genomic evidence, as explained under hypothesis H1 above.

Public genomic involvement is increasingly becoming recognized, with emphasis on the need to educate and consult the general public and those in clinical practice [52]. However, limited understanding of public engagement particularities and modalities, as well as the type of public to be involved, the methods of involving the public and the need to assess effectiveness could explain the counterproductive effect as observed in this PM implementation model. The model's analysis corroborates other research that have been carried out on public genomic awareness of recent. For instance, a recent report commissioned to investigate the public's awareness of issues around genomics in the UK, it was observed that relatively few of the public feel that they are well informed around genomics, with only one in ten (11%) stating they knew a great deal or a fair amount and a significant minority (37%) reporting they know nothing at all about this subject [53]. The report concluded that attitudes towards genomics is mixed which is unsurprising given the lack of awareness of the topic, while concerns about ethical and data protection issues raised by genomic research were equally disconcertingly divided [53]. Existing literature also point to evident mild to negative attitudes towards OBMs, and genetic testing, particularly due to anticipated emotional impact of test results, and concerns about confidentiality, stigma, and discrimination [54].

*H3. User response (UsR) due to public genomic awareness mediates the relationship between characteristic of an omics-based biomarker(OBM) and precision medicine implementation outcomes (ImO)*

The indirect effect (coefficients (a*b)) of characteristic of an omics-based biomarker(OBM) on implementation outcomes (ImO) through user response (UsR) was non-zero based on a 90% bias-corrected bootstrap confidence interval (CI = 0.030, 9.944), with a standardized point estimate of 0.073 (Table 2). This statistical evidence led to the rejection of the null hypothesis which stated that 'at 90% confidence level, the user response (UsR) due to public genomic awareness does not mediates the relationship between characteristic of an omics-based biomarker(OBM) and precision medicine implementation outcomes (ImO)'. Therefore, the alternative hypothesis was adopted.

This analysis showed that implementation outcome (ImO) increases by 0.073 standard deviations for every 1 SD increase in the characteristics of omics biomarkers (OBM) in the population indirectly via user response (UsR).

This finding is consistent with observations about holistic public genomic engagement as a crucial process in integrating genomics into healthcare systems, both at research (especially with regard to bio-sample donation) and practice settings [55]. Public acceptance of omics based biomarkers has variously been cited as a critical aspect in realizing the potential of precision medicine to improve health outcomes [56]. Additionally, issues of problematized participant consents have been resolved through genomic engagement that helps build institutional trust among the public [57].

An important question to consider in the above hypotheses testing and interpretation is whether any unmeasured and/or omitted variables might have been a basis for inferential bias. Intuitively, the superlative solution to the unmeasured variables is to reliably measure all exogenous and endogenous variables, i.e., variables that are causes of an endogenous (dependent) variable and are correlated with other causes of that endogenous variable [58]. Although measuring for the additional variables, building them into the model and statistically controlling for them would potentially be an important strategy for dealing with such confounds, it has been shown that such statistical control has a shortcoming in that it is useful only in ruling out specific, known and measurable confounds, rather than an entire class of alternative models [59]. Consequently, and in line with existing literature [58–60], we considered that in our case, measuring all potentially impactful variables for the model would be impossible to achieve. Instead we considered the operative question of the degree to which the unavoidable unmeasured variables potentially biased estimates of path coefficients and provided a basis for alternative explanations of our findings. Alternative models having to do with uncontrolled common causes may be less plausible because data analysis was conducted in a setting known to eliminate, or at least substantially reduce, the impact of important confounds (unmeasured third variables). This is because multiple indicators used to measure the latent variables for the model allowed for the modelling of correlated errors. This meant that the mediated effect, represented by coefficients 'a' and 'b', is couched in terms of error-free latent variables; thus, these values are corrected for imperfect reliability in the indicators and should be more accurate. This therefore forms the basis of the observation that based on prior knowledge, the research design used, and empirical analysis of the data used in this study, alternative models to the one presented can be ruled out.

## Conclusion

In this study, we constructed a precision medicine implementation mediated model applying SEM analysis using various r packages, including 'lavaan' and 'likert' packages. Three

hypotheses raised in the study relating to effect sizes and their significance were tested and confirmed. The relationships between characteristics of omics biomarkers (OBM) as the exogenous latent variable and user response (UsR) and implementation outcomes (ImO) as the endogenous variables were not only successfully predicted, but the mechanisms that underlie the relationship among these variables were investigated and explained. Model analysis suggests that failure or success of precision medicine implementation efforts depend on the perceived characteristics of OBMs. However, this effect is not entirely directly flowing from this perception as user response acts indirectly to influence it. The practical and theoretical implications of the intermediation as observed were discussed.

This study contributes to our understanding of the mediating processes through which precision medicine implementation outcomes are linked with perceived advantages associated with characteristics of omics biomarkers. The results have both theoretical and practice implications for biomedical genomics research and clinical genomics, respectively. For instance, the results imply better ways have to be devised to more effectively engage the public in addressing extended family support for cascade screening, especially for monogenic hereditary conditions like *BRCA*-related breast cancer and colorectal cancer in Lynch syndrome families. At basic biomedical research level, results suggest an integrated biomarker development pipeline, with early consideration of factors that may influence biomarker uptake. The results are also relevant at health systems level in indicating which factors should be addressed for successful precision medicine implementation. Admittedly however, this study had some limitations in terms of sample size, unmeasured variables and inadequate representation particularly on the gender and age aspects. Even though some of these limitations were variously mitigated in the study, there was not enough data to particularly explore and show if there were any differentiated effects across subgroups, e.g., gender and age. More data would be needed to assess moderation effects in the structural equation model.

## Acknowledgments

Authors would like to thank the University of KwaZulu-Natal, College of Health Sciences Research office for general support.

## Author Contributions

**Conceptualization:** John Jules O. Mogaka, Moses J. Chimbari.

**Data curation:** John Jules O. Mogaka.

**Formal analysis:** John Jules O. Mogaka.

**Investigation:** John Jules O. Mogaka.

**Methodology:** John Jules O. Mogaka.

**Project administration:** Moses J. Chimbari.

**Supervision:** Moses J. Chimbari.

**Validation:** John Jules O. Mogaka.

**Visualization:** John Jules O. Mogaka.

**Writing – original draft:** John Jules O. Mogaka.

**Writing – review & editing:** Moses J. Chimbari.

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
