## [Decision Letter · Decision Letter 0]

27 Apr 2020

PONE-D-19-33144

The mediating effects of public genomic knowledge in precision medicine implementation: structural equation model approach

PLOS ONE

Dear Mr Mogaka,

Thank you for submitting your manuscript to PLOS ONE. After careful consideration, we feel that it has merit but does not fully meet PLOS ONE’s publication criteria as it currently stands. Therefore, we invite you to submit a revised version of the manuscript that addresses the points raised during the review process.

We would appreciate receiving your revised manuscript by Jun 11 2020 11:59PM. To enhance the reproducibility of your results, we recommend that if applicable you deposit your laboratory protocols in protocols.io, where a protocol can be assigned its own identifier (DOI) such that it can be cited independently in the future. For instructions see: http://journals.plos.org/plosone/s/submission-guidelines#loc-laboratory-protocols

We look forward to receiving your revised manuscript.

Kind regards,

Meng-Cheng Wang

Academic Editor

PLOS ONE

Journal Requirements:

2. Your ethics statement must appear in the Methods section of your manuscript. If your ethics statement is written in any section besides the Methods, please move it to the Methods section and delete it from any other section. Please also ensure that your ethics statement is included in your manuscript, as the ethics section of your online submission will not be published alongside your manuscript.

Reviewers' comments:

Reviewer's Responses to Questions

**Comments to the Author**

1. Is the manuscript technically sound, and do the data support the conclusions?

Reviewer #1: Partly

Reviewer #2: Partly

Reviewer #3: No

2. Has the statistical analysis been performed appropriately and rigorously? 

Reviewer #1: No

Reviewer #2: Yes

Reviewer #3: No

3. Have the authors made all data underlying the findings in their manuscript fully available?

Reviewer #1: Yes

Reviewer #2: Yes

Reviewer #3: Yes

4. Is the manuscript presented in an intelligible fashion and written in standard English?

Reviewer #1: Yes

Reviewer #2: Yes

Reviewer #3: Yes

5. Review Comments to the Author

Reviewer #1: (No Response)

Reviewer #2: This study examined how characteristics of omics biomarkers influence precision medicine implementation through public genomic awareness, and clarified the relationship between several variables by structural equation modeling approach. The results have a certain value on biomedical genomics research and clinical genomics. For further modification , I have the following suggestions.

1. In the abstract, "Results from the individual mediation paths ‘a’ and ‘b’ however, showed that these effects were negative". Besause your model is not presented in the abstract, my suggestion is to explain what ‘a’ and ‘b’ refered to.

2. In order to understand the quantitative relationship between variables more clearly, is it possible to label the values in the model diagram?

3. The interpretation of the hypothetical results is slightly weak, it is recommended to further explore the results of the hypotheses.

Reviewer #3: The manuscript “The mediating effects of public genomic knowledge in precision medicine implementation: structural equation model approach” investigated the relationships among characteristics of omics biomarkers, the precision medicine implementation factor, and the public genomic awareness, leveraging structural equation modeling. The research manuscript appropriates an intriguing attempt to explore a mediation relationship, but the study has a number of weaknesses.

1. A major issue with the study is the exclusion of other significant relevant factors associated with implementation outcomes. It is often seen that two factors are significantly associated, but when including other factors associated with the dependent variable, the relationship diminished or disappeared. A more though lit review needs to be provided and including other important variables related to implementation outcomes in the model is necessary.

2. Relatedly, the authors discussed factors at different levels (e.g. innovation-level factors, individual-level factors, and organizational factors, etc.), but failed to include such factors in the quantitative model. Excluding such factors will bias the model results. Accordingly, multi-level structural equation modeling should be considered.

3. The conclusion section failed to fully explain the cause for the relationship identified through the model. How is it tied to the literature? What are the other important factors?

4. Participants’ demographic information should be included in the model. In particular, is there any differentiated effects across subgroups, e.g., gender and age? Moderation effects need to be examined in the structural equation modeling. The authors may follow this example:

Jones, M. H., Audley-Piotrowski, S. R., & Kiefer, S. M. (2012). Relationships among adolescents' perceptions of friends' behaviors, academic self-concept, and math performance. Journal of Educational Psychology, 104(1), 19–31. https://doi.org/10.1037/a0025596

5. The authors failed to explain the high attrition rate in the study (i.e. only 270 out of 442 participants completed the survey), and this may bias the study results. For example, were the participants who completed the surveys representative of the population that the authors intended to measure?

6. The authors should pay more attention to the details of the manuscript. For example, there were inconsistent spacing thought the manuscript, e.g. reference 13. There were also some missing texts, e.g. page 11 “systemic amyloidosis ().”

In summary, this study is interesting in its intention to understand precision medicine, but a more solid study design and analysis should be executed.

6. PLOS authors have the option to publish the peer review history of their article (what does this mean?). If published, this will include your full peer review and any attached files.

Reviewer #1: None

Reviewer #2: No

Reviewer #3: No

---

## [Author Response · Author response to Decision Letter 0]

21 Jun 2020

Reviewer Comments Authors’ Response Sections where changes are made in Revised Manuscript

Reviewer # 1 

1. The manuscript did follow the regular scientific writing format. For example, we do not use “research methodology” headings, we use “methods”; we do not use “statistical considerations” headings, we use “data analysis”. The authors did not have participants and procedure sessions.

 Answer: 

 “Research methodology” subtitle changed to read “Methods” as suggested;

“Statistical considerations” subtitle changed to read “Data analysis”

Although presented implicitly in a narrative form in the previous draft, the “participants and procedure” section is now more explicitly presented as a subtitle in the revised draft manuscript. 

Line 183

Line 253

From lines 184 to 218

2. The authors did not provide information about normality and outliers.

 Answer: 

Information about normality and outliers for the data has now been provided.

 Lines 306 t0 316 

3. No information about the model fits of the SEM model.

 Answer: 

Information about the model fits of the SEM model has now been provided.

 Lines 241 to 244

Reviewer # 2 

1. In the abstract, "Results from the individual mediation paths ‘a’ and ‘b’ however, showed that these effects were negative". Because your model is not presented in the abstract, my suggestion is to explain what ‘a’ and ‘b’ referred to. Answer: 

The Abstract now contains a brief statement explaining what ‘a’ and ‘b’ refer to. Lines 26 to 30

2. In order to understand the quantitative relationship between variables more clearly, is it possible to label the values in the model diagram? Answer: 

The model diagram now has the mediation coefficients labelled as suggested

 Figure 5

3. The interpretation of the hypothetical results is slightly weak; it is recommended to further explore the results of the hypotheses. Answer: 

We have now added sections on our hypotheses interpretation section to further explore the results of the hypotheses and show how findings tie up with existing literature Lines 447 t0 485

Lines 1509 to 520

Lines 542 t0 563

Reviewer # 3 

1. A major issue with the study is the exclusion of other significant relevant factors associated with implementation outcomes. It is often seen that two factors are significantly associated, but when including other factors associated with the dependent variable, the relationship diminished or disappeared. A more though lit review needs to be provided and including other important variables related to implementation outcomes in the model is necessary

 Answer: 

We believe this is a genuine concern in that the existence of unmeasured variables reflect a violation of important assumptions in path analysis. In response, and in recognizing that measuring all potentially impactful variables for the model would be impossible to achieve – as has been pointed out in related literature - we considered the question of the degree to which the unavoidable unmeasured variables potentially biased estimates of our path coefficients and provided a basis for alternative explanations of our findings. Consequently, in the revised draft, we have focused on why the presented model would likely be the most appropriate and argued out the plausibility of an alternative model based on any omitted unmeasured variables. 

Lines 542 t0 563

2. Relatedly, the authors discussed factors at different levels (e.g. innovation-level factors, individual-level factors, and organizational factors, etc.), but failed to include such factors in the quantitative model. Excluding such factors will bias the model results. Accordingly, multi-level structural equation modeling should be considered. Answer: 

In addressing this concern, we have added a section on multilevel SEM in the revised draft. We have shown that even though factors hypothesized to affect PM implementation in our study may seem to be premised on a multilevel framework , based on the available data, a multi-level structural equation modeling would not have been applicable. This is in agreement with other published works (e.g. Sophia Rabe-Hesketh, Anders Skrondal and Xiaohui Zheng, 2007(Multilevel Structural Equation Modeling) and Bengt O Muthen, 2011 (Mean and covariance structure analysis of hierarchical data)) that indicate that whereas in conventional structural equation models, all latent variables and indicators vary between units and are assumed to be independent across units, the latter assumption is violated in multilevel settings where units are nested in clusters, leading to within-cluster dependence. This is further addressed in the revised manuscript Lines 286 t0 294

3. The conclusion section failed to fully explain the cause for the relationship identified through the model. How is it tied to the literature? What are the other important factors? Answer: 

This is indeed related to Reviewer #2’s Observation #3. We have accordingly added sections on our hypotheses interpretation section to further explore the results of the hypotheses and show how findings tie up with existing literature. More specifically, we considered the question of the degree to which the unmeasured variables could have potentially biased estimates of path coefficients and provided a basis for alternative explanations of our findings. We showed how, on the basis of prior knowledge, research design used, and empirical analysis of the data used in the study, alternative models to the one presented had to be ruled out. Lines 543 t0 564

4. Participants’ demographic information should be included in the model. In particular, is there any differentiated effects across subgroups, e.g., gender and age? Moderation effects need to be examined in the structural equation modeling. Answer:

We did not do an analysis by gender and age because we did not have good representation of those groups. This is because we regrettably realized that our data does not meet important condition/assumptions for a moderated mediation: sub-groups must have similar group sizes because path coefficients depend upon subgroup sizes. Due to this consideration, we did not consider moderation as suggested. We have consequently noted this as a study limitation in our study.

 Lines 588 - 590

5. The authors failed to explain the high attrition rate in the study (i.e. only 270 out of 442 participants completed the survey), and this may bias the study results. For example, were the participants who completed the surveys representative of the population that the authors intended to measure? Answer: 

Online means of data collection are prone to increased risk of survey attrition—participants dropping out. To address this risk and possibly curb the potential havoc such participant attrition may have on our study findings, we set crucial question settings on the online survey tool to ‘compulsory’. This implied that participants had to answer the first survey question to proceed to the next one and all questions had to be completed before successfully submitting the survey form at the finish line. However, the survey platform still recorded any attempt to participate, completed or not. This measure meant that attempt by quitters to participate was registered. Unfortunately, this measure did not discriminate between genuine quitters and subsequent attempters. To eliminate possibility of a bias due to survey attrition or attrition affecting study findings and thus having a negative statistical implication on our model, we only used data from participants who actually completed the survey for all statistical analyses. Lines 206 t0 218

6. The authors should pay more attention to the details of the manuscript. For example, there were inconsistent spacing thought the manuscript, e.g. reference 13. There were also some missing texts, e.g. page 11 “systemic amyloidosis ().” Answer: 

We have proof-read the manuscripts and attended to all the formatting issues including the referencing 

 Line 76

---

## [Decision Letter · Decision Letter 1]

9 Sep 2020

PONE-D-19-33144R1

The mediating effects of public genomic knowledge in precision medicine implementation: A structural equation model approach

PLOS ONE

Dear Dr. Mogaka,

Thank you for submitting your manuscript to PLOS ONE. After careful consideration, we feel that it has merit but does not fully meet PLOS ONE’s publication criteria as it currently stands. Therefore, we invite you to submit a revised version of the manuscript that addresses the points raised during the review process.

We look forward to receiving your revised manuscript.

Kind regards,

Meng-Cheng Wang

Academic Editor

PLOS ONE

Additional Editor Comments (if provided):

I have now received the reviews of your manuscript. You will see from the accompanying comments that the reviewers recommended to accept for publication. However, they also identified several minor revisions the paper needs to be accepted for publication. Because I believe that these issues raised can be addressed easily in a revision, I encourage you to consider revising and resubmitting your paper.

Reviewers' comments:

Reviewer's Responses to Questions

**Comments to the Author**

1. If the authors have adequately addressed your comments raised in a previous round of review and you feel that this manuscript is now acceptable for publication, you may indicate that here to bypass the “Comments to the Author” section, enter your conflict of interest statement in the “Confidential to Editor” section, and submit your "Accept" recommendation.

Reviewer #2: All comments have been addressed

Reviewer #4: All comments have been addressed

2. Is the manuscript technically sound, and do the data support the conclusions?

Reviewer #2: Yes

Reviewer #4: Yes

3. Has the statistical analysis been performed appropriately and rigorously? 

Reviewer #2: Yes

Reviewer #4: No

4. Have the authors made all data underlying the findings in their manuscript fully available?

Reviewer #2: Yes

Reviewer #4: Yes

5. Is the manuscript presented in an intelligible fashion and written in standard English?

Reviewer #2: Yes

Reviewer #4: Yes

6. Review Comments to the Author

Reviewer #2: Thanks for your revision. Precision medicine emphasizes predictive, preventive and personalized treatment on

the basis of information gleaned from personal genetic and environmental data. Its

implementation at health systems level is regarded as multifactorial, involving variables

associated with omics technologies, public genomic awareness and adoption

tendencies for new medical technologies. Based on the gap in literature the authors conducted the current study. The revision work is suffient and I would like to its acceptance by the editor. This work is very interesting and has a potential to contribute in the literature.

Reviewer #4: 1.Need to report the average age , education background, family economic status and other basic demographic information about the subjects.

2. The reliability and validity of measurement tools need to be reported in manuscripts.

3. The measurement consistency coefficient of the items must to be reported.

4. Please introduce the software and version for data analysis.

7. PLOS authors have the option to publish the peer review history of their article (what does this mean?). If published, this will include your full peer review and any attached files.

Reviewer #2: No

Reviewer #4: No

---

## [Author Response · Author response to Decision Letter 1]

24 Sep 2020

Thanks for your time in reviewing this paper. I await response

---

## [Editor Report · Decision Letter 2]

30 Sep 2020

The mediating effects of public genomic knowledge in precision medicine implementation: a structural equation model approach

PONE-D-19-33144R2

Dear Dr. Mogaka,

We’re pleased to inform you that your manuscript has been judged scientifically suitable for publication and will be formally accepted for publication once it meets all outstanding technical requirements.

Kind regards,

Meng-Cheng Wang

Academic Editor

PLOS ONE
---

## [Editor Report · Acceptance letter]

2 Oct 2020

PONE-D-19-33144R2 

The mediating effects of public genomic knowledge in precision medicine implementation: a structural equation model approach 

Dear Dr. Mogaka:

I'm pleased to inform you that your manuscript has been deemed suitable for publication in PLOS ONE. Congratulations! Your manuscript is now with our production department. 

Kind regards, 

on behalf of

Dr. Meng-Cheng Wang 

Academic Editor

PLOS ONE